# People with Primary Progressive Multiple Sclerosis Have a Lower Number of Central Memory T Cells and HLA-DR^+^ Tregs

**DOI:** 10.3390/cells12030439

**Published:** 2023-01-29

**Authors:** João Canto-Gomes, Sara Da Silva-Ferreira, Carolina S. Silva, Daniela Boleixa, Ana Martins da Silva, Inés González-Suárez, João J. Cerqueira, Margarida Correia-Neves, Claudia Nobrega

**Affiliations:** 1Life and Health Sciences Research Institute, School of Medicine, University of Minho, 4710-057 Braga, Portugal; 2ICVS/3B’s, PT Government Associate Laboratory, 4710-057 Braga, Portugal; 3Division of Infectious Diseases, Center for Molecular Medicine, Department of Medicine Solna, Karolinska Institutet, 17176 Stockholm, Sweden; 4Porto University Hospital Center, 4099-001 Porto, Portugal; 5Multidisciplinary Unit for Biomedical Research (UMIB)—ICBAS, University of Porto, 4050-346 Porto, Portugal; 6University Hospital Complex of Vigo, 36312 Vigo, Spain; 7Álvaro Cunqueiro Hospital, 36312 Vigo, Spain; 8Hospital of Braga, 4710-243 Braga, Portugal; 9Clinical Academic Centre, Hospital of Braga, 4710-243 Braga, Portugal

**Keywords:** primary progressive multiple sclerosis, T cell, regulatory T cell (Treg), NK cell, NKT cell, B cell

## Abstract

The importance of circulating immune cells to primary progressive multiple sclerosis (PPMS) pathophysiology is still controversial because most immunotherapies were shown to be ineffective in treating people with PPMS (pwPPMS). Yet, although controversial, data exist describing peripheral immune system alterations in pwPPMS. This study aims to investigate which alterations might be present in pwPPMS free of disease-modifying drugs (DMD) in comparison to age- and sex-matched healthy controls. A multicentric cross-sectional study was performed using 23 pwPPMS and 23 healthy controls. The phenotype of conventional CD4^+^ and CD8^+^ T cells, regulatory T cells (Tregs), B cells, natural killer (NK) T cells and NK cells was assessed. Lower numbers of central memory CD4^+^ and CD8^+^ T cells and activated HLA-DR^+^ Tregs were observed in pwPPMS. Regarding NK and NKT cells, pwPPMS presented higher percentages of CD56^dim^CD57^+^ NK cells expressing NKp46 and of NKT cells expressing KIR2DL2/3 and NKp30. Higher disease severity scores and an increasing time since diagnosis was correlated with lower numbers of inhibitory NK cells subsets. Our findings contribute to reinforcing the hypotheses that alterations in peripheral immune cells are present in pwPPMS and that changes in NK cell populations are the strongest correlate of disease severity.

## 1. Introduction

Multiple sclerosis (MS) is a chronic disorder from the central nervous system (CNS) characterized by the occurrence of brain lesions and demyelinating plaques due to immune-mediated myelin damage [1,2]. Lesions at the CNS can occur throughout the brain and spinal cord and lead to symptoms that range from motor to cognitive impairments. The MS symptoms, type and rate of disease progression can greatly vary among patients, which hampers and delays diagnosis. Because of the heterogeneous nature of MS, some individuals already present a high degree of disability when receiving the definitive diagnosis of MS [3]. Two main forms of MS are defined according to its progression: relapsing remitting MS (RRMS) and primary progressive MS (PPMS). Most people with MS (pwMS) suffer from RRMS, which is characterized by episodes of disability (relapses), followed by periods of partial to total recovery (remission). PPMS affects about 10–15% of pwMS and refers to a state of the steady accumulation of neurological disability, independent of relapses, since the onset of the disease clinical manifestation [4].

Inflammation of the CNS, demyelination and neurodegeneration are common pathophysiological features of all MS forms. Yet, whether those processes start in the periphery or in the CNS, and whether the immune players that contribute to those features are the same between different MS forms, is still unknown. More is known about the involvement of immune cells in the pathophysiology of RRMS, as compared to PPMS. This is due to the availability of mouse models that recapitulate RRMS and due to the demonstrated clinical efficacy of therapies targeting T and B cells on RRMS progression. In fact, the generalized lack of effect of most DMDs available for RRMS on pwPPMS leads to the hypothesis that while RRMS depends on events implicating the peripheral immune system, the events involved in PPMS might not be as dependent on the periphery but rather might be mostly circumscribed to the CNS. Still, the involvement of the peripheral immune system in PPMS pathophysiology cannot yet be excluded, as the treatment of pwPPMS with natalizumab, a monoclonal antibody that binds α4-integrin and hampers the transmigration to the CNS of T- and other VLA-4 expressing cells, reduces brain tissue damage and markers of intrathecal inflammation in the cerebrospinal fluid (CSF) [5,6]; these changes are accompanied by a slight tendency toward decreased disease severity [7]. Additionally, the only approved DMD for PPMS, Ocrelizumab, targets CD20^+^ cells, proving the involvement of B cells in disease progression; in the Oratorio clinical trial, it was shown that continuous treatment provided sustained benefits regarding measures of disease progression through the 6.5 years of follow-up [8]. B cells are proposed to contribute to PPMS either as cytokine and antibody producers or as antigen-presenting cells essential for T cell activation [9]. Additionally, B cells have been proposed to function as a reservoir for the Epstein–Barr virus, one of the strongest MS risk factors [10].

Regarding T cells, studies report alterations in treatment-naïve pwPPMS compared to healthy controls (HC), including reduced thymic export [11,12] and alterations on the memory T cell subsets, such as a lower number of effector memory CD4^+^ T cells or a lower percentage of both naïve and effector memory CD8^+^ T cells [13,14]. Although potentially controversial, these studies support the presence of peripheral immune system alterations in pwPPMS. Discrepancies between those studies could possibly be explained by the merging of pwPPMS and people with secondary progressive MS (pwSPMS; a progressive form that is the late stage of RRMS in most patients) and/or the lack of a proper experimental control for relevant variables known to impact the immune system such as age, sex and human cytomegalovirus (HCMV) seroprevalence [11,12,15]. Most of those studies focus on the analysis of conventional CD4^+^ and CD8^+^ T cells and regulatory cells, including regulatory T cells (Tregs); Natural Killer (NK) cells and NKT cells have been seldom explored in pwPPMS and might have a role in disease pathophysiology. Importantly, regulatory cells not only have an essential role in inducing tolerance to self-molecules (including proteins and/or lipids/glycolipids) but also contribute to oligodendrocyte progenitor cell differentiation and remyelination in mouse models of MS [16,17]. 

Previous data from our group highlighted new and important immune features in T cells and regulatory cells (Tregs, NK and NKT) in pwRRMS [18]. Yet, some of those phenotypes of immune cells are barely explored in PPMS. In this study, we focused on pwPPMS to evaluate which immune alterations are present in treatment-naïve pwPPMS compared to age- and sex-matched HC. To achieve this goal, a broad characterization of peripheral T cell subsets, including conventional CD4^+^ and CD8^+^ and Tregs, as well as subsets of B cells, NKT cells and NK cells in pwPPMS and HC, is performed. This will add to the better understanding of the immune cells alterations that might underlie PPMS pathophysiology and might provide markers of disease progression.

## 2. Results

### 2.1. pwPPMS Present a Lower Number of Circulating Central Memory CD4^+^ T Cells

To better clarify which peripheral immune system alterations might underlie PPMS, we evaluated the percentage and number of the main immune subsets among CD4^+^ and CD8^+^ T cells. No differences in the percentage and number of circulating CD4^+^ T cells between pwPPMS and HC were observed (Figure 1A, Appendix A). Based on the expression of CD45RA and CCR7, the naïve and memory subsets were defined as follows: naïve (CD45RA^+^CCR7^+^), central memory (CM; CD45RA^−^CCR7^+^), effector memory (EM; CD45RA^−^CCR7^−^) and terminally differentiated CD45RA-expressing memory T cells (TemRA; CD45RA^+^CCR7^−^; Figure 1B). Within the naïve subset, the recent thymic emigrants (RTEs) were defined as the cells expressing CD31 (Figure 1C). RTEs’ number and percentage were not different between pwPPMS and HC (Figure 1D), nor was the naïve-to-memory ratio (Figure 1E). Comparable percentages and numbers of naïve CD4^+^ T cells (Figure 1F) and similar percentages of memory CD4^+^ T cells were observed between both groups, though pwPPMS had a lower number of total memory cells (Figure 1G). No major differences were observed regarding the number and percentage of CM, EM and TemRA CD4^+^ T cells, apart from the number of CM, which was lower in pwPPMS (Figure 1H–J). The proliferation rate of total CD4^+^ T cells, as assessed by Ki67 expression, was similar between pwPPMS and HC (Figure 1K).

The number and percentage of total CD8^+^ T cells and their naïve and memory subsets were also evaluated (Appendix A). We found a tendency for a lower number of CM CD8^+^ T cells and a significantly higher percentage of EM CD8^+^ T cells in pwPPMS (Appendix A). Upon controlling for age and sex in the multiple linear regression model, the tendency found on CM cells became statistically significant (Appendix A). A tendency toward an increased percentage of proliferating CD8^+^ T cells was found in pwPPMS compared to HC (Appendix A).

In addition to T cells, the percentages and numbers of CD20^+^ B cells subsets were evaluated. No differences were found in the percentage or number of total CD20^+^ B cells (Appendix A). Based on the expression of CD27, the naïve (CD27^−^) and memory (CD27^+^) B cells were defined (Appendix A). No differences were found between pwPPMS and HC regarding the naïve, memory or naïve-to-memory ratio among CD20^+^ B cells (Appendix A). Yet, a tendency toward a lower percentage of naïve and a lower naïve-to-memory ratio in PPMS was found after controlling for age and sex in multiple linear regression models (Appendix A).

Altogether, our data suggest that pwPPMS present a contraction in the number of peripheral CM T cells in both CD4^+^ and CD8^+^ T cell compartments and no alterations on B cells.

### 2.2. pwPPMS Display a Lower Number of Activated HLA-DR^+^ among TREGS 

Tregs are poorly characterized in PPMS, and their contribution to disease progression remains undetermined. No differences between pwPPMS and HC regarding the percentage and number of total Tregs were observed (Figure 2A, Appendix A), nor in the percentage of proliferating Tregs (Figure 2B). The expression levels of HLA-DR have been described to positively correlate with FOXP3 expression levels, which determines the suppressive capacity of Tregs [19,20]. Three Tregs subsets were defined based on the expression of CD45RA and HLA-DR: naïve (CD45RA^+^HLA-DR^−^), activated Tregs expressing HLA-DR (CD45RA^−^HLA-DR^+^) and activated Tregs lacking HLA-DR expression (CD45RA^−^HLA-DR^−^; Figure 2C). When comparing pwPPMS and HC, no differences were observed in the naïve-to-activated-Tregs’ ratio, nor in the naïve, activated HLA-DR^−^ or activated HLA-DR^+^ Tregs percentages and cell numbers, except for the number of activated HLA-DR^+^ Tregs (Figure 2D–G). In pwPPMS, a tendency toward a lower number of activated HLA-DR^+^ Tregs was found. This tendency became statistically significant in the multiple linear regression model upon controlling for age and sex (Figure 2G, Appendix A). In summary, pwPPMS were characterized by a lower number of activated HLA-DR^+^ Tregs, a subset described to present the highest gene expression of molecules related with Tregs cytotoxic function and contact-dependent activation [21].

### 2.3. pwPPMS Have a Higher Percentage of the Most Mature NK Cells Expressing NKp46 and of NKT Cells Expressing KIR2DL2/3 and/or NKp30 

The number and phenotype of NK cells were evaluated in pwPPMS and HC. The NK cell subsets studied were defined according to CD56 and CD57 expression as follows: the most immature CD56^bright^ cells, the intermediate CD56^dim^CD57^−^ and the most differentiated subset CD56^dim^CD57^+^. Additionally, an evaluation of the expression of inhibitory (KLRG1, NKG2A, KIR2DL2/3 and KIR3DL1) and activating (NKp30, NKp40 and NKp46) receptors was performed within all NK cell subsets [22]. pwPPMS did not differ from HC in terms of the number or percentage of total NK cells or any of its subpopulations (Appendix A). Regarding the percentage of NK cells expressing each of the inhibitory receptors KLRG1, NKG2A or KIR3DL1, pwPPMS did not differ from HC (Appendix A). Despite pwPPMS presenting a significant increase in the percentage of total NK and CD56^dim^CD57^+^ NK cells expressing KIR2DL2/3, the multiple linear regression models for these populations were not statistically significant (Appendix A). No differences were found in the expression of the activating NKp30 or NKp44 receptors in any NK cell subset between pwPPMS and HC (Appendix A). Notwithstanding, pwPPMS had a higher percentage of CD56^dim^CD57^+^ NK cells expressing NKp46 (Appendix A). No differences were found regarding the number of NK cell subsets expressing any of the activating or inhibitory receptors evaluated (data not shown).

Concerning NKT cells, no differences were observed in their total percentage or number (Figure 3A, Appendix A). pwPPMS presented higher percentages of NKT cells expressing KIR2DL2/3 or KLRG1 and higher numbers of NKT cells expressing KIRDL2/3; no differences were found for NKG2A- or KIR3DL1-expressing NKT cells (Figure 3B–E, Appendix A). Concerning the activating receptors (Figure 3F–H, Appendix A), pwPPMS presented a higher percentage and number of NKT cells expressing NKp30. No differences in NKT cells expressing NKp44 were observed between pwPPMS and HC. Despite the higher percentage of NKT cells expressing NKp46 in pwPPMS that was observed, the multiple linear regression model for this population was not significant (Figure 3G,H, Appendix A).

Taken together, these results show that, in comparison to HC, pwPPMS present a higher percentage of the most differentiated NK cell subset expressing NKp46 and a higher number of NKT cells expressing KIR3DL2/3 and/or NKp30.

### 2.4. Disease Severity and Time from Diagnosis Are Mostly Correlated with Alterations on NK Cells Subsets

To investigate whether severity and years since disease diagnosis correlate with the immune system’s cells alterations, we performed linear regression models considering the percentage or number of each cell population previously described as a dependent variable and as an independent variable—either disease severity or time since diagnosis (Appendix A).

MS severity relies on patients’ symptoms and is often measured using the expanded disability status scale (EDSS): a higher disease severity results in a higher EDSS, which ranges from 0, when the neurologic exam is normal, to 10, when there is death due to MS. We found that the EDSS relates specifically to alterations on NK cells (Figure 4A, Appendix A). Higher values of EDSS were related to a lower percentage of total NK cells and to lower number of total NK cells expressing NKG2A. Additionally, higher values of EDSS were related to lower numbers of CD56^dim^CD57^−^ NK cells, specifically the ones expressing the inhibitory receptors KIR2DL2/3, KLRG1 and/or NKG2A and the activating receptor NKp46. NK cell numbers also correlated with the time since diagnosis (Figure 4B, Appendix A). People living longer with diagnosed PPMS had a lower number of the most immature CD56^bright^ NK cells; a lower number of total NK cells and CD56^dim^CD57^−^ NK cells expressing the inhibitory receptor NKG2A was also related to a longer time since diagnosis. Time since PPMS diagnosis is also negatively correlated with the number of activated HLA-DR^+^ Tregs (Figure 4B). No other cell populations correlated with EDSS or with time since diagnosis on the linear regression models.

Altogether, the clinical parameters EDSS and years since diagnosis were related to alterations in the percentage and number of immune cells, particularly in NK cells. Importantly, those alterations point towards pwPPMS having fewer NK cells expressing inhibitory receptors with increasing severity and years from diagnosis.

## 3. Discussion

In the present work, a lower number of memory CD4^+^ T cells, specifically CM cells, was observed in pwPPMS. Our results do not recapitulate a reduced thymic export and premature immune aging in pwMS, as was observed by others [11,23]. In our study, we included only pwPPMS with no history of any disease-modifying drug treatment, and for the control group, we considered only age- and sex-matched healthy individuals. In addition, we also performed multiple linear regressions including age and sex variables to exclude their contribution to the immune system cells’ alterations. These regression models were revealed to be essential in uncovering the immune system alterations associated with PPMS that otherwise would not have been detected, as is the case for the number of CM CD8^+^ T cells and activated HLA-DR^+^ Tregs and for the percentage of KLRG1^+^ and NKp46^+^ cells among NKT cells. HCMV infection also has a strong impact on immune cell populations [24]. In our cohort, only two individuals were HCMV IgG seronegative (one HC and one pwPPMS), and for this reason, this variable was not included in the multiple linear regression models. 

Our study supports the idea that pwPPMS have fewer memory cells than HC, specifically CM CD4^+^ and CD8^+^ T cells. Interestingly, we have previously shown that newly diagnosed pwRRMS also have fewer memory cells, particularly EM CD8^+^ T cells [18]. The different findings made in pwPPMS and pwRRMS can reflect the involvement of distinct memory T cell subsets in different forms of MS and/or at different disease stages (newly diagnosed pwRRMS vs. pwPPMS with a median time since diagnosis of 1 vs. 26 months). Teniente-Sierra et al. observed that pwPPMS have fewer early EM cells among both CD4^+^ and CD8^+^ T cells but saw no differences in the CM cells numbers [13]. These early EM cells succeed CM T cells. Although in a different memory subset, those results corroborate the reduced number of memory T cells in pwPPMS. Differences in the markers used to discriminate T cell subsets (i.e., inclusion of CD27) and the merging of pwPPMS and pwSPMS in a single group might explain the differences found in distinct memory T cell subsets. 

The involvement of B cells in PPMS has been highlighted by the efficacy of therapies specifically targeting CD20^+^ B cells. Importantly, CD20 is not expressed by plasma B cells, the B cells that produce antibodies. The mechanism by which CD20^+^ B cells depletion leads to clinical improvement therefore cannot be exclusively through antibodies production. In fact, after CD20^+^ depletion therapies, no short-term measurable differences in the antibody’s titers are observed, although clinical improvement [25]. A tendency was found to lower naïve-to-memory ratio accompanied by a lower percentage of naïve B cells in pwPPMS after controlling for age and sex. Several studies found alterations in naïve and memory B cells subsets in MS [9]: a higher percentage of naïve B cells expressing CD86 and CCR5 in untreated pwRRMS; increased levels of B cells expressing TNFα, IFNy, IL12 and IL6 in pwPPMS and reduced regulatory B cells expressing IL13, IL10 and TGFβ in pwPPMS compared to HC [26,27]. Importantly, it has been shown that higher values of EDSS correlate positively with the percentage of naïve cells but negatively with the percentage of memory B cells in pwRRMS [28]. Yet, here, no correlation was found between EDSS or time since diagnosis with B cells subsets in pwPPMS. Nevertheless, the knowledge on the phenotype and function of B cells subsets in PPMS is still incomplete. A more extensive analysis of B cells’ phenotype and function with increased sample sizes in pwPPMS is needed to better understand the PPMS pathophysiology. 

The activated HLA-DR^+^ Tregs subset is described to be more cytotoxic, as they induce contact-dependent suppression [21]. It was previously reported that pwMS present a higher expression of HLA-DR among activated Tregs and a higher percentage of activated Tregs and a lower percentage of naïve Tregs [29]. We could not observe differences in Tregs subsets percentages, but a lower number of HLA-DR^+^ Tregs was found in pwPPMS. Importantly, the number of activated HLA-DR^+^ Tregs decreased with the time since diagnosis, suggesting that a longer time living with the disease correlates with a progressive decrease in the number of contact-dependent suppressive Tregs.

NK and NKT cells have a very important role in immune homeostasis, as they express inhibitory and/or activating receptors that allow these cells to identify potential hazardous target cells and kill them, including activated autoreactive T cells [30,31]. The most immature NK cells, CD56^bright^, are described to act mostly through cytokine production, while CD56^dim^ mediate their cytotoxicity in a contact-dependent manner. The most differentiated and cytotoxic NK cells are the CD56^dim^CD57^+^. Laroni et al. showed that the blocking of NKp30 and NKp46 receptors in CD56^bright^ NK cells from healthy individuals hampered their capacity to suppress autologous T cell proliferation and induce their death [32]. In addition, they observed that autologous T cells from untreated pwMS circumvent CD56^bright^ NK cells suppression by upregulating HLA-E, an NKG2A ligand that inhibits NK cells’ direct cytotoxicity [33]. The CD56^bright^/CD56^dim^ ratio and the surface phenotype and intracellular content of the granzyme B of CD56^bright^ NK cells are described to be similar between pwMS and HC [32]. Additionally, in vitro studies also suggested that CD56^dim^ NK cells from pwMS are not essential to suppressing autologous T cells, as no significant differences in death and proliferation were observed on autologous T cells when cultured in the presence or absence of CD56^dim^ NK cells [32]. Similar to Laroni et. al.’s findings, we found no differences in the phenotype of CD56^bright^ NK cells between pwPPMS and HC. However, an increasing time from diagnosis was related to decreasing numbers of that NK cell subset, specifically the ones expressing NKp44. Additionally, we found that pwPPMS presented a higher percentage of CD56^dim^CD57^+^ NK cells expressing NKp46. Although very few differences were found in NK cells between pwPPMS and HC, this was the cell subset that related the most with the disease severity and the time since disease diagnosis. We observed that the number of CD56^dim^CD57^−^ NK cells expressing the inhibitory receptors KIR2DL2/3, KLRG1 and/or NKG2A was inversely correlated with disease severity. This suggests that pwPPMS with higher disease severity scores present CD56^dim^CD57^−^ NK cells that are less prone to be inhibited. 

The T cell receptor on NKT cells recognizes lipids and glycolipids in the context of the CD1d molecule [31,34,35]. In experimental autoimmune encephalomyelitis, an MS mouse model, the lack of NKT cells led to a severe disease phenotype [36]; in addition, the stimulation of NKT cells in wild-type mice and the adoptive transfer of NKT cells to mice lacking these cells inhibited Th1 and Th17 responses in vivo, protecting mice from severe disease [35,36]. Those studies suggest a role for NKT cells in inducing tolerance in mice; yet, data on NKT cells in humans are still scarce. We found that pwPPMS present a higher percentage and number of NKT cells expressing the KIR2DL2/3 and/or NKp30 receptors. Importantly, NKp30 binds to Galectin-3 (Gal-3) [37], which has been found in mouse models to be important for the maturation and ramification of oligodendrocytes, the myelin-producing cells in the CNS [38,39]. Interestingly, in vitro studies showed that culturing oligodendrocyte progenitor cells (OPCs) isolated from rats with CSF from pwPPMS induced morphological changes and an aberrant transcriptional program in those cells [40]. Importantly, a higher expression of the gene encoding for Gal-3 was confirmed in normal-appearing white matter from the *post-mortem* brain tissue of pwPPMS. The higher expression of Gal-3 by the oligodendrocytes in pwPPMS might render those cells more prone to being recognized by NKT cells expressing NKp30, which we found to be increased in the peripheral blood of pwPPMS. 

The small sample size might represent a limitation of this study, particularly on the regression models assessing the relation between the immune cell subsets and the clinical parameters studied. Moreover, this study is focused on alterations in the peripheral immune cells; studies comparing these alterations with the ones observed in the CSF and/or brain lesions would be valuable in understanding to what extent these peripheral alterations are mirrored in the two compartments. In addition, functional studies are required to further understand whether these phenotypic alterations impact the proper function of the immune system in pwPPMS vs. HC.

## 4. Conclusions

pwPPMS are characterized by lower numbers of CM CD4^+^ and CD8^+^ T cells and of the most suppressive Treg subset and a higher percentage of NKT cells expressing inhibitory and activating receptors. Moreover, the time since disease diagnosis and, more evidently, the disease severity are variables negatively related with the number of NK cells expressing inhibitory receptors. Altogether, these results point toward pwPPMS having fewer memory T cells and phenotypical alterations on regulatory cells that might affect their suppressive potential.

## 5. Materials and Methods

### 5.1. Study Population

The participants were recruited between 2018 and 2022 at Hospital de Braga (Braga, Portugal), Hospital Geral de Santo António (Centro Hospitalar Universitário do Porto, Porto, Portugal) and Hospital Álvaro Cunqueiro (Vigo, Spain). The inclusion criteria included: diagnosis of PPMS; older than 18 years old; and naïve for MS disease-modifying drugs. Participants were excluded whenever they presented: (i) other autoimmune or immunodeficiency diseases; (ii) a history of radiotherapy and/or chemotherapy; (iii) treatment with corticosteroids in the last 3 months or continuously for more than 6 months; (iv) splenectomy or thymectomy; and/or (v) pregnancy. Clinical and demographic data were recovered from medical records (Table 1). For the control group, samples from healthy individuals recruited in the context of the current study or that of the Switchbox cohort study were used [41].

### 5.2. Blood Processing

Blood samples from pwPPMS and HC were collected and processed as previously reported [18]. Briefly, blood was collected into K_2_EDTA tubes and blood cells’ counts evaluated by FACS staining either using the MUSE cell analyzer or the BD LSRII analyzer (using counting beads). Peripheral blood mononuclear cells (PBMCs) and plasma were isolated from whole blood using Histopaque-1077 (Sigma-Aldrich, Dorset, UK) gradient centrifugation. Plasma was stored at −80 °C in 2 mL aliquots. PBMCs were stored into aliquots of 5 million cells in RPMI 1640 media supplemented with 20% FBS (both from PanBiotech, Aidenbach, Germany) and 10% dimethyl sulfoxide (DMSO; Sigma-Aldrich). Samples were kept at −80 °C on an isopropanol containing MrFrosty for at least 24 h and transferred to liquid nitrogen afterwards. 

### 5.3. Anti-Human Cytomegalovirus IgG seroprevalence

Plasma IgG antibodies against the human cytomegalovirus (HCMV) were semi-quantified through the anti-HCMV IgG Elisa kit (Abcam, Cambridge, UK), according to the manufacturer’s instructions.

### 5.4. FACS Staining

Briefly, PBMCs aliquots were thawed and divided to be stained with four distinct antibody panels to characterize CD4^+^ and CD8^+^ T cells’ naïve and memory subsets, Tregs and their activation status and NK/NKT cells and their expression of activation and inhibitory receptors (Appendix A). The CD4^+^ and CD8^+^ T cells’ naïve and memory panel were acquired on the same day from fresh cells. Tregs and NK/NKT cells panels were acquired the next day from fixed cells. 

Samples were acquired using FACS Diva Software v6.0 (Becton Dickinson, East Rutherford, NJ, USA) on a BD LSRII flow cytometer. Rainbow calibration beads (BioLegend, San Diego, CA, USA) were used to standardize the cytometry voltage settings across experiments. To control for experimental and analysis bias, an internal control was used; this control consisted of cells from the same individual, isolated and stored prior to the beginning of the study, which was always thawed and stained in each assay together with the patients’ samples. FlowJo Software v10 (Becton Dickinson) was used to analyze the data; gating strategies are represented in Appendix A. Cell subsets from a parent population were evaluated only when the parent population had at least 500 events.

### 5.5. Statistical Analysis

The statistical analysis was performed using IBM SPSS Statistics v26 (IBM Corporation, Armonk, NY, USA) and GraphPad Prism v9 (GraphPad Software, San Diego, CA, USA). Differences were considered statistically significant for *p*-values < 0.050. The normal distribution of the data was evaluated through the skewness and kurtosis measures and the D’Agostino & Pearson test. For data following a normal distribution, an independent Student’s t-test was performed to compare between two independent groups. For data following a non-normal distribution, a Mann–Whitney U-test was used to compare two independent groups. A comparison of the qualitative data (i.e., the percentage of males or anti-HCMV IgG^+^ individuals) between two groups was performed through the Chi-square test. The effect size (magnitude of difference) was calculated as in [18]: for the t-test, the measure of the effect size Cohen’s d was considered small, medium or large when d < 0.300, [0.300,0.800] or >0.800, respectively; for the Mann–Whitney U-test, the measure of the effect size r was considered small, medium or large when r < 0.300, [0.300,0.500] or >0.500, respectively. The sample size, statistical outputs and effect size are described in Appendix A. To further control for the impact of age and sex on the immune cell populations evaluated, multiple linear regression models were performed. Either the percentage or number of each immune cell population was included as a dependent variable in regression models; age, sex (reference category: female) and PPMS (reference: healthy) were included as independent variables. To meet the assumption of normality for the multiple linear regression models, dependent variables not following a normal distribution were log10-transformed. The impact of the time since diagnosis or the disease severity (EDSS) on the immune cell populations alterations was assessed through a sequential multiple linear regression followed by a linear regression using the residuals of the former, as previously described [18,42]. Briefly, to discard the effect of sex and age on immune cells alterations, the unstandardized residuals were retrieved from multiple linear regression models having, as a dependent variable, the percentage or number of each cell population and, as independent variables, age and sex. Those residuals (i.e., the variation in the cell population not explained by the independent variables of age and sex) were then used as dependent variables on subsequent linear regression models having, as independent variables, EDSS or time since PPMS diagnosis to obtain the predictive value of each clinical parameter. 

## Figures and Tables

**Figure 1 cells-12-00439-f001:**
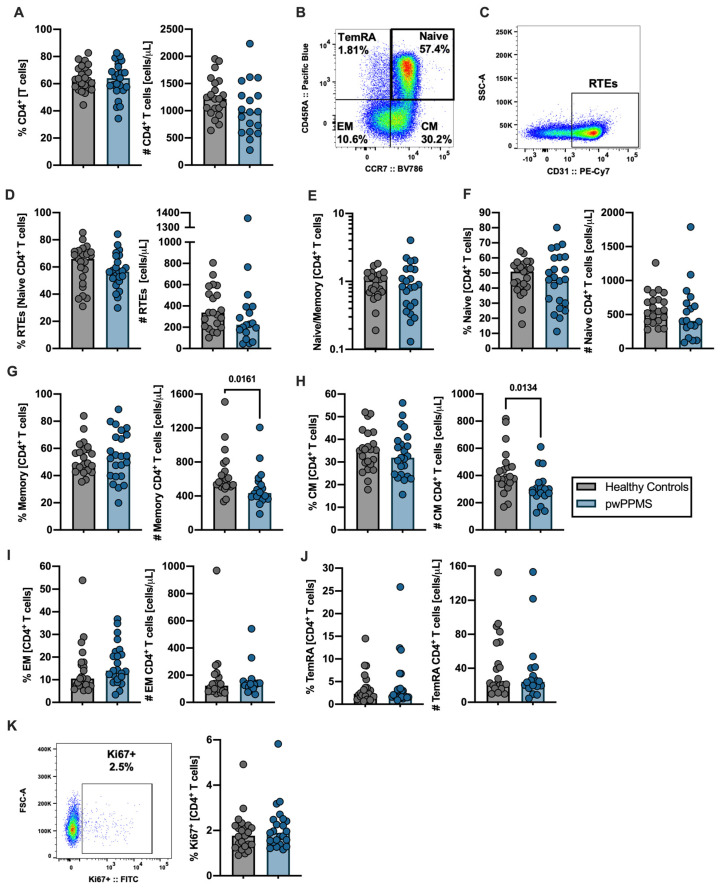
People with PPMS present lower levels of central memory CD4^+^ T cells. The number and percentage of CD4^+^ T cells are represented for healthy controls (gray circles) and pwPPMS (blue circles; (**A**)). Representative dot plots for the identification of CD4^+^ T cell subpopulations based on the expression of CD45RA and CCR7: naïve (CD45RA^+^CCR7^+^); central memory (CM; CD45RA^−^CCR7^+^), effector memory (EM; CD45RA^−^CCR7^−^); and terminally differentiated CD45RA-expressing memory T cells (TemRA; CD45RA^+^CCR7^−^; (**B**)). Representative dot plots of recent thymic emigrants (RTEs), as assessed by the expression of CD31 within the naïve CD4^+^ T cells population (**C**). Representation of the percentage and number of RTEs (**D**). CD4^+^ T cells’ naïve-to-memory ratio, where total memory CD4^+^ T cells were considered as the sum of CM, EM and TemRA subsets (**E**). Representation of the percentage and number of CD4^+^ T cell subsets: naïve (**F**), total memory (**G**), CM (**H**), EM (**I**) and TemRA (**J**). Representative dot plot and percentage of proliferating CD4^+^ T cells (Ki67^+^; (**K**)). All graphs display a dot per individual, with the horizontal lines representing the groups’ median. The parametric Student’s *t*-test was performed in ((**A**,**D**); percentage), ((**F**,**G**); percentage) and (**H**); the non-parametric Mann–Whitney U-test was performed in ((**D**); number), (**E**), ((**F**,**G**); number) and (**I**,**J**,**K**). *p*-values were represented in the graph whenever they were <0.100, and differences were considered statistically significant at a *p*-value < 0.050. Statistical outputs and effect size calculations are presented in Appendix A; multiple linear regression model outputs for assessing the impact of sex and age on the CD4^+^ T cell subpopulations variations are depicted in Appendix A.

**Figure 2 cells-12-00439-f002:**
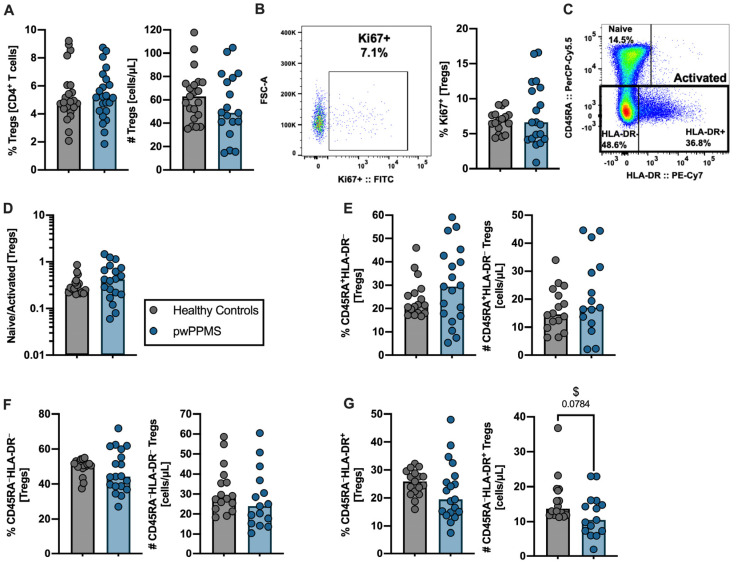
A lower number of activated CD45RA-HLA-DR^+^ Tregs are observed in pwPPMS. The percentage and number of Tregs (CD3^+^CD4^+^CD127^low^CD25^+^FoxP3^+^) are displayed for healthy controls (gray circles) and pwPPMS (blue circles; (**A**)). Representative dot plot and percentage of proliferating (Ki67^+^) Tregs (**B**). Tregs subpopulations were defined based on CD45RA and HLA-DR expression according to representative dot plots: naïve (CD45RA^+^HLA-DR^−^); activated HLA-DR^−^ (CD45RA^−^HLA-DR^−^); and activated HLA-DR^+^ (CD45RA^−^HLA-DR^+^; (**C**)). Ratio of naïve to activated Tregs, where the total activated Tregs were considered as the sum of activated HLA-DR^−^and activated HLA-DR^+^ subsets (**D**). Percentage and number of the Tregs subsets: naïve (**E**), activated HLA-DR^−^ (CD45RA^−^HLA-DR^−^; (**F**)) and activated HLA-DR^+^ (CD45RA^−^HLA-DR^+^; (**G**)). All graphs display a dot per individual, with the horizontal lines representing the groups’ median. The parametric Student’s *t*-test was performed in (**A**,**B**), ((**E**,**F**); number) and ((**G**); percentage); the non-parametric Mann–Whitney U-test was performed in (**D**), ((**E**,**F**); percentage) and ((**G**); number). *p*-values were represented in the graph whenever they were <0.100, and differences were considered statistically significant for a *p*-value < 0.050. Statistical outputs and effect size calculations are presented in Appendix A. $ stands for situations when significance is gained upon controlling for age and sex on the multiple linear regression models (Appendix A).

**Figure 3 cells-12-00439-f003:**
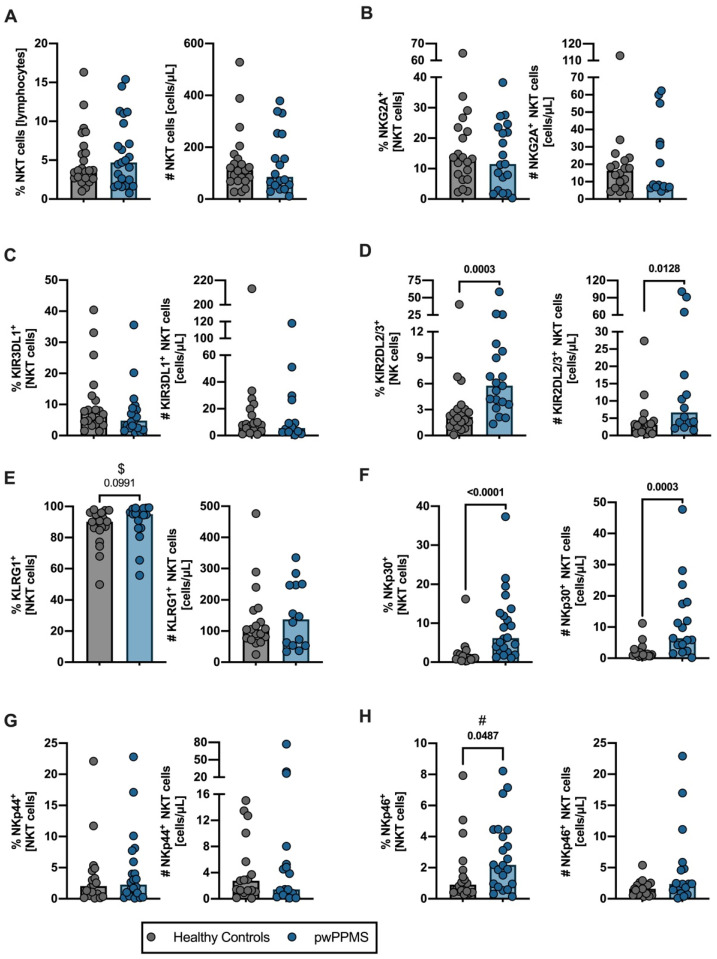
People with PPMS display a higher percentage of NKT cells expressing KIR2DL2/3 and/or NKp30. The percentage and number of NKT (CD3^+^CD56^+^) cells are represented for healthy controls (gray circles) and pwPPMS (blue circles; (**A**)). Representation of the percentage and number of NKT cells expressing inhibitory receptors: NKG2A (**B**), KIR3DL1 (**C**), KIR2DL2/3 (**D**) and KLRG1 (**E**). Percentage and number of NKT cells expressing activating receptors NKp30 (**F**), NKp44 (**G**) and NKp46 (**H**). All graphs display a dot per individual, with the horizontal lines representing the groups’ median. All comparisons were performed using the non-parametric Mann–Whitney U-test. *p*-values were represented in the graph whenever they were <0.100, and differences were considered statistically significant for a *p*-value < 0.050. Statistical outputs and effect size calculations are presented in Appendix A. $ and # stand for situations when significance is gained or lost, respectively, upon controlling for age and sex on the multiple linear regression models (Appendix A).

**Figure 4 cells-12-00439-f004:**
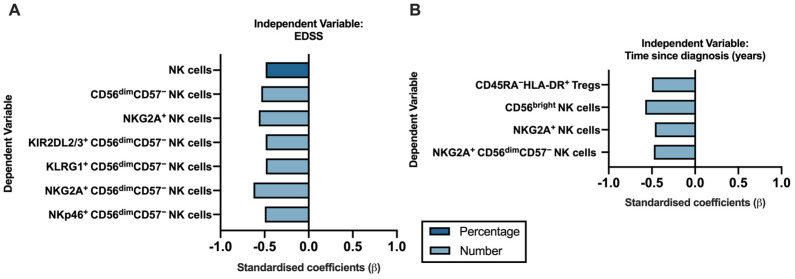
A longer time since diagnosis and a higher disease severity negatively correlate with NK cell subsets alterations on pwPPMS. The standardized coefficients (beta) from the linear regression models are represented, having as independent variables either the expanded disability status scale (EDSS; (**A**)) or the time since diagnosis (**B**). The dependent variable consisted of the unstandardized residuals from a preliminary multiple linear regression where the contribution of sex and age on the blood populations was evaluated (as described in the Materials and Methods section). Only the blood populations that yield a significant linear regression model for the variable of interest are represented. The linear regression model’s output for EDSS and the time since diagnosis are represented in Appendix A, respectively.

**Table 1 cells-12-00439-t001:** Demographic and clinical characterization of the cohort.

	pwPPMS (*n* = 23)	Healthy Controls (*n* = 23)
Age (years). Median [range]	54.7 [24;74]	54.4 [25;76] ^1^
Men. % (*n*)	30.4 (7)	30.4 (7) ^2^
Age at PPMS onset (years). Median [range]	50 [24;73]	na
Time since MS diagnosis (years). Median [range]	2.2 [0.2;22.6]	na
EDSS. Median [range]	5 [2;7]	na
anti-HCMV IgG^+^. % (*n*)	95.7 (22)	95.7 (22) ^3^
^1^ t44 = −0.095; *p* = 0.9240		
^2^ χ2 (df = 1) = 0.000; *p* = 1.000		
^3^ χ2 (df = 1) = 0.000; *p*= 1.000		

HCMV, Human Cytomegalovirus; EDSS, Expanded Disability Status Scale; na, not applicable.

## Data Availability

The original data presented in the study are included in the article/Appendix A. Further inquiries can be directly addressed to the corresponding author.

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
