# Peer review of "People with Primary Progressive Multiple Sclerosis Have a Lower Number of Central Memory T Cells and HLA-DR+ Tregs"

_cells, 2023, doi:10.3390/cells12030439_

Round 1
Reviewer 1 Report
The authors present new data on specific immune cell subpopulations - T and regulatory population (Tregs and NK cells). The comparisons are made against healthy controls and associations with clinical data are presented. The methods are presented clearly.
There is room for improving the manuscript:
1) Although B-cells are mentioned briefly in the introduction were analysis on naive and memory B cell populations performed. This data will improve the findings. Currently, it reads half complete.
2) More needs to be added to the introduction to present the reasoning behind the work performed. For example, why focus on PPMS and what is different between RRMS and PPMS in terms of immune cell composition; the make up of factors that directly indirectly lead to autoimmunity.
3) There are no limitations to the study in the discussion this needs to be included. For example, the sample size is small particularly for the clinical associations.
Reviewer 2 Report
The role of peripheral immune responses in PPMS is controversial. Here, Canto-Gomes and colleagues assess differences in the lymphocyte (T, NK, NKT) populations and markers in the peripheral blood of pwPPMS vs controls (HC). Data are presented both uncorrected, as well as corrected for sex and age. They detected perturbations in pwPPMS among central memory CD4+ and CD8+, effector memory CD8+, HLA-DR+ Treg, NKp46 expression on differentiated NK cells, and activation/inhibition markers on NKT cells. Interestingly, lower NK/NKT subsets correlate with higher EDSS and longer time spent living with PPMS.
On the whole, I believe that this well powered analysis of lymphocyte subsets from patient-derived samples is an interesting and important contribution to the field. Of note, the (unfortunate) paucity of treatments for PPMS allows the investigators to draw conclusions without the confounding variable of treatment regimen, which is a complicating factor in RRMS.
There are, however, some issues that need correction/clarification and these are outlined below.
* in the introduction/discussion, there is a lack of description of B cells and their potential role. I can accept that this is not the focus of the analysis, but the possible role of B cells ought to be discussed when one considers that anti-CD20 ocrelizumab is one of the few DMTs to show any effectiveness in PPMS (ORATORIO study)
* in Fig 1K, representative Ki67 stainings ought to be presented
* in Fig 2 legend, it would be helpful to have the basic Treg phenotype (CD4+CD127(low)FoxP3+CD25+, I think?) written out, so that the reader does not have to dig into the Supplementary Figures to ascertain the gating strategy. Ditto for NK and NKT in the respective main figures
* Figure 3 is labeled 2x
* The 2nd Figure 3 (ie Fig 4), are there p-values that can be presented for each correlative analysis?
* In "Fig 4" (correlative analysis), were multiple comparisons controlled for?
* the manuscript is well written, but there are minor grammatical issues that can be improved. Ie first line of intro, "Multiple sclerosis is *a* chronic disorder..."
Round 2
Reviewer 1 Report
The additional changes have greatly improved the manuscript. There are only minor typos in the new edits; in the discussion section 'titres' (line 314) and 'positively' (line 321) need to be corrected.